# Sorption Isotherms, Glass Transition and Bioactive Compounds of Ingredients Enriched with Soluble Fibre from Orange Pomace

**DOI:** 10.3390/foods11223615

**Published:** 2022-11-12

**Authors:** Claudia Perez-Pirotto, Gemma Moraga, Isabel Hernando, Sonia Cozzano, Patricia Arcia

**Affiliations:** 1Departamento de Ingeniería, Universidad Católica del Uruguay, Montevideo 11600, Uruguay; 2Food Microstructure and Chemistry Research Group, Department of Food Technology, Universitat Politècnica de València, 46022 Valencia, Spain; 3Latitud LATU Foundation, Montevideo 11500, Uruguay

**Keywords:** citrus byproducts, powder stability, critical water activity, critical water content, physical properties, antioxidant capacity

## Abstract

Citrus fruits are one of the main crops worldwide. Its industrialization, primarily juice production, produces large amounts of byproducts, composed of seeds and peels, that can be used to obtain new ingredients. In this study, sorption behaviour, glass transition, mechanical properties, colour and bioactives of four different soluble fibre-enriched powders obtained from orange pomace using green technologies were studied. Powders were equilibrated at water activities between 0.113 and 0.680 for fifteen weeks at 20 °C, and studies were performed to indicate the best storing conditions to ensure the glassy state of the amorphous matrix and higher bioactive stability. By combining the Gordon and Taylor model with the Henderson isotherm, the critical water activity and content for storage in a glassy state were determined. The ingredient obtained after extrusion + hot water is the most stable, which is also the one with the highest dietary fibre content. Powder obtained by jet cooking is the least stable, as it is not in a glassy state at any water activity at room temperature. To increase storage stability, these should be stored at refrigeration temperatures.

## 1. Introduction

In recent decades, the interest in consuming foods with high content of dietary fibre has increased, as it has been linked to a lower prevalence of certain chronic diseases [1,2]. Following this trend, new sources of fibre are being investigated [2,3]. Although cereals have a higher amount of dietary fibre, fruits’ fibre has a better profile, with a higher share of soluble fibre. Citrus fruits are of special interest, as their fibre has associated bioactive compounds, such as flavonoids and polyphenols [4]. 

Because of this particularity of citrus fruits, work has been conducted with orange peel waste, a by-product of orange juice production [4,5,6]. As only 50% of fresh fruit is used in juice production, and orange is one of the most cultivated fruits around the world, the amount of pomace produced is very big [7,8]. Traditionally, it has been used as animal feed or for pectin extraction, which is performed with acids [9,10]. Moreover, in line with current trends following green chemistry [11], research has been carried out to improve the fibre profile of the orange pomace, with environmentally friendly technologies (without solvent use). Some of the assayed technologies are high hydrostatic pressures [12], steam explosion [13] and ultrasound-assisted extraction [14]. Work has been also been carried out with extrusion, hot water and steam use through the use of jet cooking [15,16,17,18,19]. After treatments, soluble fibre can be extracted and dried to obtain a powder ingredient that can be used in the formulation of other foods [18].

A characteristic of fruits and ingredients derived from them is the high content of simple carbohydrates in their composition. As these have low molecular weight, their glass transition temperature (T_g_) is low. Therefore, at room temperature, the matrix could be in a rubbery state, leading to stickiness, caking and agglomeration in products [20,21,22]. Furthermore, these matrixes are very sensitive to small variations in temperature and moisture content, which are common in product storage [23,24]. Hence, to prevent these changes that lead to losses in product quality, the matrix should be kept in a glassy state.

To predict physical, chemical, and microbiological stability changes, the knowledge of sorption isotherms may be very useful. These relate the water activity (a_w_) of the sample with its water content (x_w_), at a given temperature. Therefore, sorption isotherms are useful to determine the optimum conditions for storage, to preserve shelf life and for choosing appropriate packaging materials [25,26]. They are dependent on the structure and composition of the material, as well as temperature and pressure [20,27].

By combining the isotherms with glass transition temperatures, the modified state diagram of the amorphous phase can be represented (T_g_-x_w_-a_w_ diagram). With it, a critical value of water content (CWC) and water activity (CWA) can be determined, depending on storage temperature. These values are important factors for the stability of the product [23].

Several works studying this relationship have been carried out. Flores-Ramírez et al. (2022) [28] studied the effect of adding maltodextrin in T_g_-x_w_-a_w_ relationship in freeze-dried juices. Work has also been performed with dried honey and the effects of different carrier materials [20]. Al-Ghamdi et al. (2020) [25] studied the a_w_-x_w_-T_g_ relationship in pumpkin, together with its colour stability. González et al. (2020) [29] studied this relationship in dried persimmon, and also the effect of glass transition in colour and mechanical properties. This relationship has also been linked to chemical and physical properties, such as powder flowability or caking, as Cruz-Tirado et al. (2021) [30] studied in their work. 

In this scenario, the aim of this work was to study the a_w_-x_w_-T_g_ relationship for a fibre-enriched ingredient derived from orange pomace, obtained using novel extraction technologies. We intended to determine the effect of glass transition on the mechanical properties and bioactive compounds during storage.

## 2. Materials and Methods

### 2.1. Sample Preparation

Orange fibre powders were obtained according to Perez-Pirotto et al. (2022) [18]. Four different treatments (hot water extraction—HW; extrusion + hot water extraction—EHW; jet cooker—JC; jet cooker + hot water extraction—JCHW) were assayed on orange pomace to obtain a soluble fibre-enriched powder. 

For extrusion, pomace was dried in a convection oven for 72 h and milled to a particle size < 1.0 mm. Sample was extruded following Huang and Ma (2016) [17], with the following optimum conditions: moisture content of the sample 15%, 230 rpm and 129 °C in the three zones of a single-screw extruder. After extruding, sample was ground again and a hot water extraction was performed, mixing it with water (1:16.6 *m/v*) and keeping it at 75 °C for 45 min under continuous agitation. 

Hot water, jet cooking and jet cooking + hot water were performed on wet orange pomace after being ground to a particle size < 1.0 mm. Hot water treatment was performed by mixing pomace with water (1:2 *m/v*) and kept at 75 °C for 45 min under continuous agitation. In jet cooking, sample was mixed with water (1:2.5 *m/v*) and the slurry was passed with a pump through the heating element, with a residence time between 3 and 5 s. Temperature was 85 °C and steam pressure was 2.8 bar, while exit pressure was 1.0 bar. In jet cooking + hot water, the sample was heated at 75 °C for 45 min after coming out of the jet cooker, in a mechanically agitated boiler. 

After recovering the supernatant from each treatment, it was concentrated under vacuum and spray-dried on a spray drier (Buchi B290) with whey protein isolate as encapsulating agent (8% soluble solids). Drying conditions were 130 °C inlet air temperature, atomization air flow 414 L/h, liquid feed pump rate 35 m^3^/h, and outlet temperature 70 ± 4 °C. Powders were kept in laminated bags at room temperature until use. The composition of these powders has been previously studied by Perez-Pirotto et al. (2022) [18]. Briefly, results per 100 g of dry sample were: total dietary fibre (TDF) 9.48 ± 0.45 g in HW, 20.20 ± 3.69 g in EHW, 10.25 ± 0.89 g in JC and 13.03 ± 1.8 in JCHW; total sugars content 64.15 ± 0.2 g in HW, 46.15 ± 0.05 g in EHW, 64.24 ± 0.13 g in JC and 60.37 ± 0.31 g in JCHW.

About 10 g of each sample was placed in aluminium cups and stored in the dark in airtight containers with saturated saline solutions (LiCl, CH_3_COOK, MgCl_2_, K_2_CO_4_, Mg(NO_3_)_2_ and KI), with water activities (a_w_) of 0.112, 0.230, 0.330, 0.430, 0.520, and 0.680, respectively [31]. After fifteen weeks of storage, equilibrium moisture content was determined by weight difference [28]. Initial water content was determined by vacuum drying in a vacuum oven at 60 °C until constant weight (Vaciotem, JP Selecta SA, Abrera, Spain).

### 2.2. Sorption isotherm 

Water content data were fitted to Henderson, Caurie and BET models. Nonlinear regression analysis was determined by statistical software Solver (Excel 2016). These models are predicted by Equations (1) to (3), respectively.
(1)we=0.01[−log(1−aw)10f]1n
where w_e_ is the equilibrium water content (g water/g solids), a_w_ is the water activity and f and n are characteristic parameters of the product.
(2)$$w_{e}=\exp\left[a_{w}\cdot \ln\left(r\right)-\frac{1}{4.5\cdot w_{s}}\right]$$
where w_e_ is the equilibrium water content (g water/g solids), a_w_ is the water activity, r is a characteristic constant of the material and w_s_ is the moisture security (g water/g solids) content that gives the highest stability to dehydrated product during storage.
(3)$$w_{e}=\frac{w_{o}{\;C\;a}_{w}}{\left(1-a_{w}\right)\left(1+\left(C-1\right)a_{w}\right)}$$
where w_e_ is the equilibrium water content (g water/g solids), a_w_ is the water activity, w_o_ is the monolayer moisture content (g water/g solids) and C is the sorption energy constant. 

### 2.3. Glass Transition Temperature

Glass transition temperature of each powder at each water activity was determined by differential scanning calorimetry. About 10 mg of sample was placed in aluminium pans (Tzero pans—TA Instruments) and analysed using a DSC25 (TA Instruments). Heating rate was 10 °C/min, and the temperature ranged from −90 °C and 70 °C, depending on the sample’s water content. Nitrogen was used as the purge gas, with a rate of 40 mL/min. Midpoint temperature was considered the glass transition temperature. 

Experimental data were fitted to Gordon and Taylor model, following Equation (4): (4)$$T_{g}=\frac{\left(1-x_{w}\right)T_{g}\left(\mathrm{as}\right)+{k\;x}_{w}{\;T}_{g}\left(w\right)}{\left(1-x_{w}\right)+k_{w}}$$

x_w_ is the mass fraction of water (g water/g product), T_g_ is the glass transition temperature (°C), T_g_(w) the glass transition temperature of amorphous water (−135 °C), T_g_(as) the glass transition temperature of anhydrous solids predicted by the model and k is the model constant. 

### 2.4. Colour Analysis

Colour parameters were measured with a CR-400 Colorimeter (Minolta Co., Ltd., Osaka, Japan), using D65 illuminant/10° observer to obtain CIE L*a*b* colour coordinates. Total colour difference (ΔE*) was calculated by Equation (5). Three samples per treatment per chamber were analysed.
(5)$$\Delta E^{*}=\sqrt{{\left(L^{*}-L_{0}^{*}\right)}^{2}+{\left(a^{*}-a_{0}^{*}\right)}^{2}{\left(b^{*}-b_{0}^{*}\right)}^{2}}$$
where L* is the luminosity and a* and b* the intensities of the green-red colour and the blue-yellow colour, respectively. The variables without subscript correspond to the equilibrated samples, while the others are from the lowest water sample (a_w_ = 0.113) [30]. 

Chroma (C*) and hue angle (h*) were also calculated with Equations (6) and (7).
(6)$$C^{*}=\sqrt{a^{*2}+b^{*2}}$$
(7)$$h^{*}=\tan^{-1}\frac{b^{*}}{a^{*}}$$

### 2.5. Texture Analysis 

A texture analyser (TA.XT2, Stable Micro Systems, Surrey, England), was used to measure hardness of the equilibrated samples. Using the compression mode, the force (N) required to penetrate the sample was measured. A 4 mm diameter, flat-tipped cylindrical probe was used. A penetration distance of 20 mm and descendent rate of 10 mm/s were used. Samples were placed in cylindrical cups and three samples per treatment per chamber were analysed. 

### 2.6. Bioactive Characterization

Bioactive compounds were extracted with ethanol. Briefly, about 100 mg of sample were placed in centrifuge tubes, and 1.0 mL of ethanol 96% was added. Samples were vortexed and centrifuged (Eppendorf centrifuge 5415 R) at 10,000 rpm for 10 min at 4 °C. After centrifugation, 0.9 mL were removed, and the same amount of fresh ethanol was added. After vortexing and centrifuging, the supernatant was combined with previously removed ethanol, and taken to a final volume of 10 mL with ethanol. Extracts were kept away from light at −18 °C until use. 

Polyphenol content was determined by Folin–Ciocalteu method, as described by [18]. Briefly, 1.0 mL of extract was mixed with 6 mL of distilled water and 0.5 mL of Folin–Ciocalteu reagent. After vortexing and 3 min in the dark, 1.0 mL of 20% Na_2_CO_3_ solution and 1.5 mL of water were added. Tubes were vortexed and left in the dark for 90 min, when absorbance was measured at 765 nm in plastic cuvettes. Gallic acid was used for calibration purposes. Antioxidant capacity was assayed using DPPH [32] and FRAP [33] modified methods., as described by [18]. For FRAP analysis, 30 µL of water and 30 µL of sample (ethanol for blank) were mixed with 900 µL of freshly prepared FRAP reagent and left at 37 °C for 30 min in the dark. Absorbance was measured at 595 nm after that time. For DPPH analysis, 1 mL of extract was mixed with 0.004% (*w/v*) freshly prepared DPPH reagent in ethanol and left in the dark for 30 min at room temperature. Control was prepared with DPPH and ethanol. Absorbance was read at 517 nm. In both antioxidant assays, Trolox was used for calibration purposes.

Three samples per treatment per chamber were analysed.

### 2.7. Statistical Analysis

One-way analysis of variance (ANOVA) was performed with XLSTAT 2022.1.2 (Addinsoft 2022, New York, NY, USA). A Tukey test was used to evaluate the significant differences (*p* > 0.05) between samples. 

## 3. Results

### 3.1. Sorption Behaviour and Glass Transition 

Three sorption isotherm models (BET, Henderson and Caurie) were fitted to the experimental data, and results are shown on Table 1.

Only experimental data up to a_w_ ≤ 0.5 were fitted to the BET model, as with higher values the model hypothesis failed. Brunauer’s classification classifies sorption isotherms in four types according to their C value, a constant related to sorption energy [34]. In HW, EHW and JC, isotherms are type III, as C < 2, which is commonly seen in products with relatively high fibre, sugar and protein content [35]. This means these powders have low water adsorption until the water activity allows for solubilization when adsorption increases. In JCHW, the isotherm is type II, which is common in sugars and crystalline materials [36]. This type indicates that the sample has an amorphous and porous structure and that the sample’s sorption consists of multilayer adsorption [20]. 

Monolayer moisture content (g water/g solids) varied from 0.094 for JCHW to 0.158 in the case of JC. This value is interesting, as it expresses the amount of water that is sufficient to cover the adsorbing surface with a layer of water molecules of the thickness of one molecule. Some authors have described it as an optimal water content for stability of some low-moisture foods, as it is a critical product storage moisture content value to maintain the physical, chemical, sensorial and microbial quality of dried foods [20,36].

Caurie and Henderson isotherms had the best fit to the data, as their R^2^ are the highest. Caurie’s w_s_ is the security water content, which has been considered as the one that would ensure the maximum stability to the dehydrated product during storage, as it related to the to the maximum water content that prevents an important increase in the rate of deteriorative reactions [37]. This parameter was lower than the monolayer water content (w_o_) predicted by the BET model in all cases. 

The model that fitted best to experimental data was the Henderson type. This model has been recommended for the description of the isotherms of polysaccharide materials [38].

The variations in the environmental humidity imply changes in powders’ water content and water activity, as modelled by the sorption isotherms. These variations can provoke the glass transition of the amorphous phase [23]. All samples showed glass transition in the analysed temperature ranges. This relation between glass transition and water content can be modelled by the Gordon and Taylor equation, as presented in Equation (5). Table 2 shows the parameters of this model for the four powders. 

Moreover, to obtain the critical water content and activity values related to glass transition, the combined T_g_-x_w_-a_w_ data and the Henderson and Gordon and Taylor fitted models were used. Results are shown in Figure 1. In the cases of HW, EHW and JCHW, powders could be stored at 20 °C, but in the case of JC at this temperature, the powder is already in its amorphous state. 

To keep the HW, EHW and JCHW powders in their glassy state at 20 °C, they must be stored in an atmosphere with moisture lower than CWA, and their water content should be lower than the CWC. In the case of HW, the maximum relative humidity of the environment that would ensure this is 8.0%, and the product would need to have a maximum water content of 0.0120 g water/g product. In the case of EHW, the maximum relative humidity that would ensure glassy state is 17.3%, with the product having a maximum 0.0315 g water/g product. In the case of JCHW, the critical water activity was 0.082, so environmental humidity should be lower than 8.2%, and product’s water content must be 0.0169 g water/g product maximum. In the three cases, CWC was found to be lower than monolayer moisture content modelled by the BET model (w_o_), which indicates that this value might fail to assure quality preservation during the storage of this type of products. This has also been seen in the case of freeze-dried grapefruits [23] and freeze-dried persimmons [29]. As reported by Roos (1993) [39], some of the problems of dehydrated foods, such as collapse and stickiness, are more related to the glass transition than to the monolayer water content. Caurie’s water security content (w_s_) was also found to be higher than CWC at 20 °C in all cases, but the difference with the predicted CWC is less pronounced than the one of the monolayer moisture content predicted by the BET model. 

At refrigeration temperatures (5 °C), it was possible to find CWC and CWA for all the powders. In this condition, powders would be allowed to have a higher water content and to be stored at higher water activities. Moreover, the storage of powders at refrigeration temperatures would slow down the rate of deteriorative reactions. 

The observed differences in T_g_ predicted by the Gordon and Taylor model can be due to the different composition of powders. As this parameter is mainly affected by water content and molecular weight, when keeping the former constant, the glass transition is expected to be decreased with lower molecular weights. According to Perez-Pirotto et al. (2022) [18], JC is the powder with the highest amount of sugars and the lowest amount of dietary fibre, which would indicate that its molecular weight is lower. The opposite happens in the case of EHW, with the highest T_g_ values. 

### 3.2. Colour Analysis

The parameters of lightness (L*), chroma (C*), hue angle (h*) and colour differences (ΔE*) are presented in Table 3. 

Lightness (L*) decreased with increasing water activity in all treatments, which indicates that samples were darker when water content was higher, as can be seen in Figure 2. Chroma (C*) values varied differently, as these were higher in water activities around 0.3 and 0.4. As the chroma value represents the saturation or purity of the colour, the obtained results show that colours were the purest around these water activities. At lower or higher values, colours tended to be less saturated. 

Hue angle values in all cases increased with increasing water activity, showing that colours tended to be more yellowish in lower water activities, and as water content increased, the colour became more reddish. By combining these results, it is concluded that when water activity increases, the powders tend to be more brownish (lower luminosity values and more reddish). This has also been seen in the case of pumpkin powder [25].

Colour changes were evaluated by the colour difference parameter (ΔE*). As all colour differences were higher than 1.5, these differences are perceptible to the human eye [40]. Non-enzymatic browning is most likely to occur in low-moisture systems, due to the concentration of solutes being higher because of the low water content. In that situation, interactions between sugars and reducing amino acids are increased [22]. However, as reported by Al-Ghamdi et al. (2020) [25], this is higher when the matrix is in the amorphous state. Therefore, the JC powder exhibited more colour difference from lower water activities, probably due to the resultant browning. 

HW, JC and JCHW have a higher colour difference from 0.520 water activity. Their water content was around 13 g/100 g sample, well above the CWC at 20 °C, but similar to the 11% that promotes the enzymatic browning reactions according to Ling et al. (2005) [41]. In these cases, the colour changes seem to be related not only to the glass transition but also to the amount of water in the sample. Taking into account that above the glass transition temperature the molecular mobility is higher and viscosity is lower, the reactions rate at higher water activities (with lower glass transition temperatures) is expected to be higher [42]. Although the colour difference is highest in all cases at a_w_ = 0.680, the greatest increase at that water activity is seen only in EHW, probably due to its highest glass transition temperature. 

### 3.3. Texture Analysis

Texture analysis results are shown on Figure 3. Lower water activities kept the powder looser, as can be seen in Figure 2. Higher water activities changed the product’s appearance; a paste was formed that was softer as water content increased. 

These changes in texture can be related to water content and crystalline state. When the environmental temperature approaches glass transition temperature, water plasticizes the material and structure changes, collapsing and becoming denser. Moreover, as water content increases, more liquid bridges between particles will be formed, increasing caking and sticking, as seen by Mosquera et al. (2011) [22] in spray-dried borojó. This is because above the glass transition temperatures, molecules are able to rearrange from the glassy state to a very viscous, liquid-like state, allowing for stickiness and viscous flow [43]. 

The rate of caking is a function of T_g_, time and relative humidity. As the temperature difference T-T_g_ (ΔT) increases, the rate of caking is faster. When relative humidity is enough to keep the powder on its glassy state, no caking is observed. When relative humidity decreases the glass transition temperature below storage temperature, the caking begins. The rate of this phenomena will be higher if relative humidity is higher. At the beginning of caking, the formation of interparticle bridges will carry an increase in the force needed to compact the sample [22]. This is what is seen in the case of JC and HW at a water activity of 0.430, while in the case of EHW and JCHW, it happens when water activity reaches 0.520. These differences may be due to the differences in their composition [18], as JC and HW have a similar content of total sugars, and the lowest total dietary fibre content. On the other hand, EHW and JCHW have lower amounts of total sugars and higher dietary fibre contents, which implies that their molecular weight is higher [18]. Telis and Martínez-Navarrete (2009) [24] found that the decrease in maximum force in texture analysis of grapefruit powder was dependent on the type of sugar added, and it happened at higher water activities when low-dextrose-equivalent maltodextrin or gum arabic were added, in comparison to high-dextrose-equivalent maltodextrin. This would indicate that in the case of higher molecular weight, the decrease in maximum force would happen at higher water activities. 

Once caking is fully developed, the force needed to compact the sample will be lower, as sample will be completely liquefied [22]. As the rate of caking is dependent on the temperature difference ΔT, it is expected that at higher water activities, where T_g_ is lower and therefore ΔT is higher, caking is fully developed, and samples are liquified [37]. These changes would be minimized if the samples were stored at refrigeration temperatures.

When relating texture changes to colour changes, it seems that mechanical properties have place before the highest colour change, as in HW and JC the highest hardness is seen at a_w_ = 0.430, and in EHW and JCHW it is seen at a_w_ = 0.520. These samples exhibit the maximum colour difference increase at a_w_= 0.520 and a_w_ = 0.680, respectively. This was also seen by Telis and Martínez-Navarrete (2010) [37]. 

### 3.4. Bioactive Characterization

The total polyphenol content and antioxidant capacity of the four samples stored at each water activity are shown in Figure 4 and Figure 5. 

Total polyphenol content and antioxidant capacity (both in FRAP and DPPH) followed the same trend: EHW was the powder with the highest polyphenol content and antioxidant capacity at all water activities. According to [18], this powder is the one with the highest dietary fibre content and may have a higher amount of polyphenols bonded to it. 

In the case of EHW, the highest polyphenol content was observed at a water activity of 0.430. This maximum value may be related to a higher extractability of phenolic compounds, as observed by Maldonado-Astudillo (2019) [44], who found that at higher water activities, phenolic compounds can be released due to the matrix swelling and dissolution. The decrease seen at higher water activities (0.520 and 0.680) may also be due to an increase in degradation of these compounds.

Regarding antioxidant capacity, the same trend is observed. However, in the case of EHW and JCHW, no significant difference was observed between the values for a_w_ 0.113 and 0.430 (*p* < 0.05). EHW and JC are the ones with the most extreme thermic treatment, which could result in the formation of melanoidins and reductones from Maillard reactions, with antioxidant capacity [18]. In HW and JC, the stability of bioactive compounds (polyphenols and antioxidants) is lower, as a decrease is observed on antioxidant capacity and total phenol content when increasing water content. This was also reported in grapefruit powder after storage by Moraga et al. (2019), who found a negative linear correlation between antioxidant capacity and relative humidity. 

In HW and JC cases, the decrease is more remarkable from a_w_ = 0.430, where products have a moisture content of near 10 g/100 g sample. Until a_w_ = 0.330, products’ moisture content (below 7 g/100 g sample) was near the w_s_ predicted by Caurie’s model. At higher water activities, moisture content is much higher than security water content, and the bioactive compounds are more affected. This decrease may be related to their lower stability, as they are the products with the lowest CWA and CWC. 

## 4. Conclusions

The results of this investigation indicate that the properties and stability of the obtained powders depend on the extraction method used. The stability parameters confirm the problem of these kind of products, in which the high specific surface that can adsorb water makes the product very hygroscopic. This leads to collapse and bioactive compound loss.

To avoid these issues, it is necessary to ensure that the matrix is in a glassy state throughout storage. However, in these cases, the critical water activity and content that would ensure this state are extremely low, being the highest in the case of extrusion + hot water treatment. In the case of the jet cooker sample, the matrix is not in a glassy state at any water activity nor with any water content at 20 °C. In the extrusion + hot water sample, on the other hand, with the highest fibre content, the changes in texture and colour and the bioactive compound loss are less than in the other samples, and it would be in its glassy state if stored with a water content below 3.15% and with water activity below 0.173. 

To increase the stability of these ingredients, especially the ones obtained by the other extraction procedures, samples should be stored at refrigeration temperatures. 

The obtention of powdered ingredients from orange pomace is a way of adding value to the citric production chain. However, investigation is needed to test the possible application in food formulation. 

## Figures and Tables

**Figure 1 foods-11-03615-f001:**
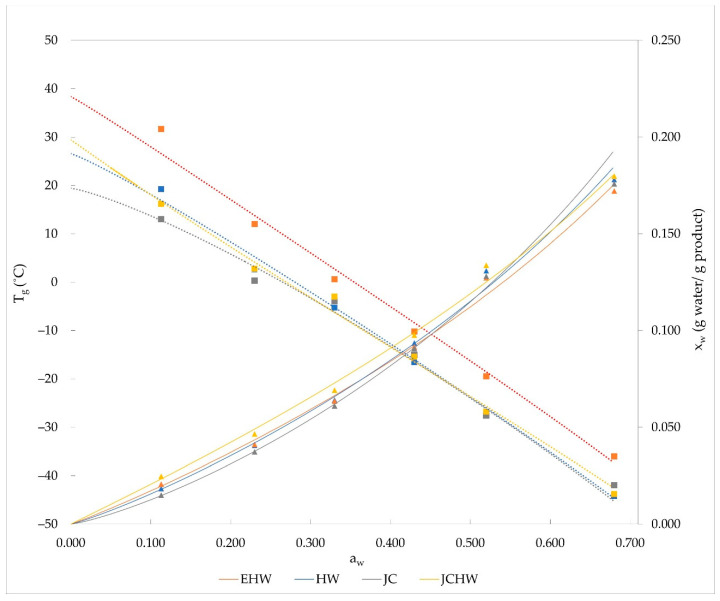
Temperature–water activity and water activity–water content relationship of different treatments. HW: hot water; EHW: extrusion + hot water; JC: jet cooker; JCHW: jet cooker + hot water. Experimental data (T_g_ ν and x_w_▲) and fitted models (Henderson and Gordon and Taylor models). Dashed line predicted with Equation (1); continuous line predicted with Equation (5).

**Figure 2 foods-11-03615-f002:**
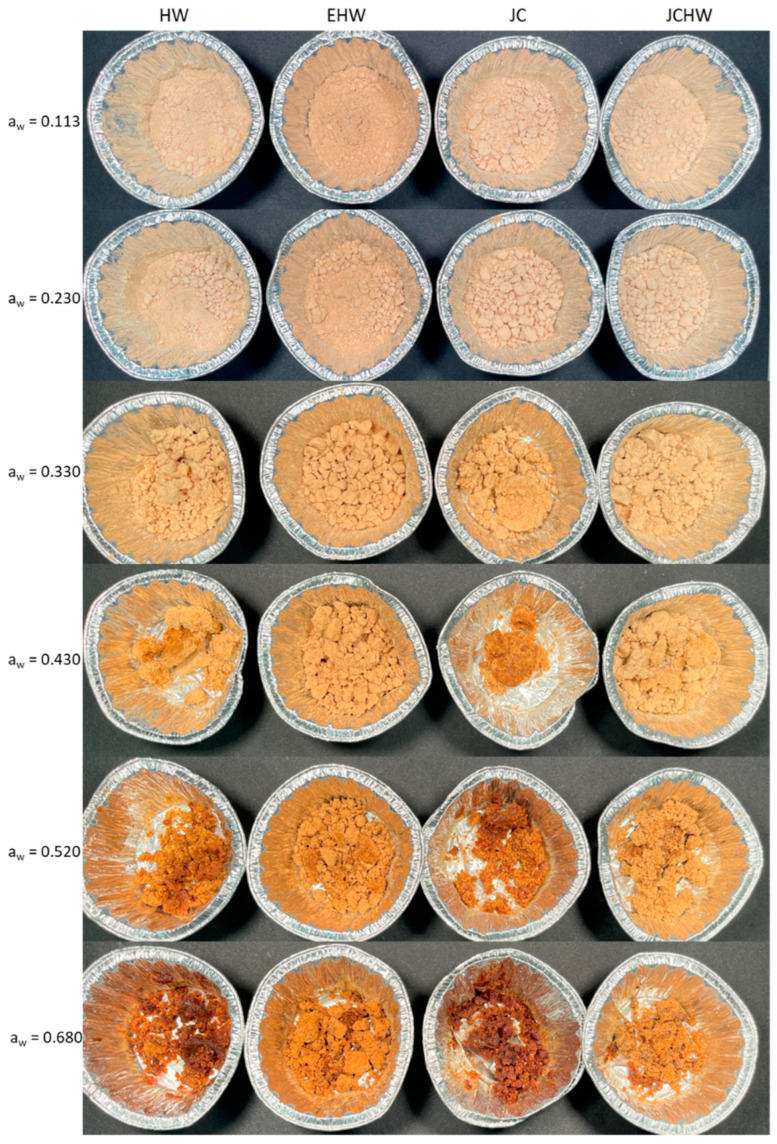
Powder photographs after three months of storage at different water activities. HW: hot water; EHW: extrusion + hot water; JC: jet cooker; JCHW: jet cooker + hot water.

**Figure 3 foods-11-03615-f003:**
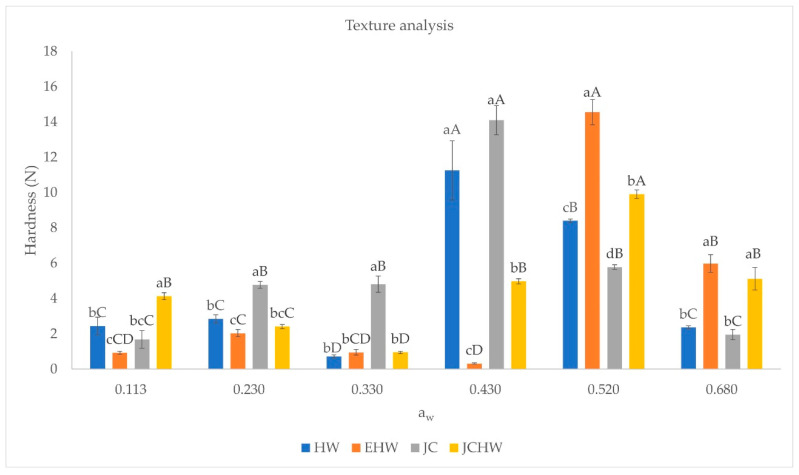
Hardness at different water activities. HW: hot water; EHW: extrusion + hot water; JC: jet cooker; JCHW: jet cooker + hot water. Different small letters for the same a_w_ indicate significant difference according to Tukey test (*p* < 0.05) between treatments. Different capital letters in the same treatment indicate significant difference according to Tukey test (*p* < 0.05) between water activities.

**Figure 4 foods-11-03615-f004:**
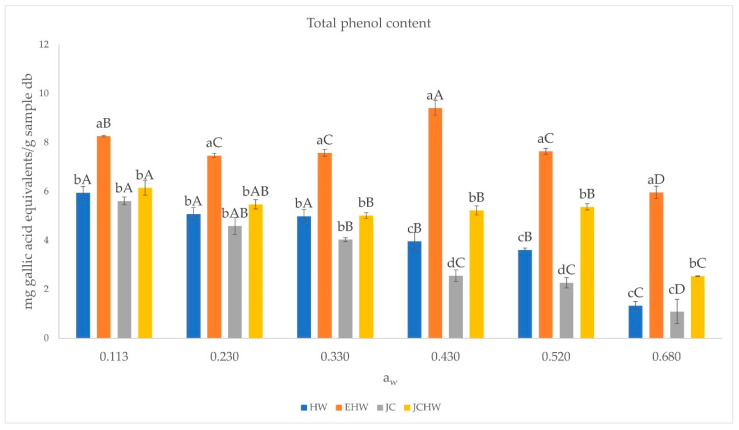
Total polyphenol content at different water activities. HW: hot water; EHW: extrusion + hot water; JC: jet cooker; JCHW: jet cooker + hot water. Different small letters for the same aw indicate significant difference according to Tukey test (*p* < 0.05) between treatments. Different capital letters in the same treatment indicate significant difference according to Tukey test (*p* < 0.05) between water activities.

**Figure 5 foods-11-03615-f005:**
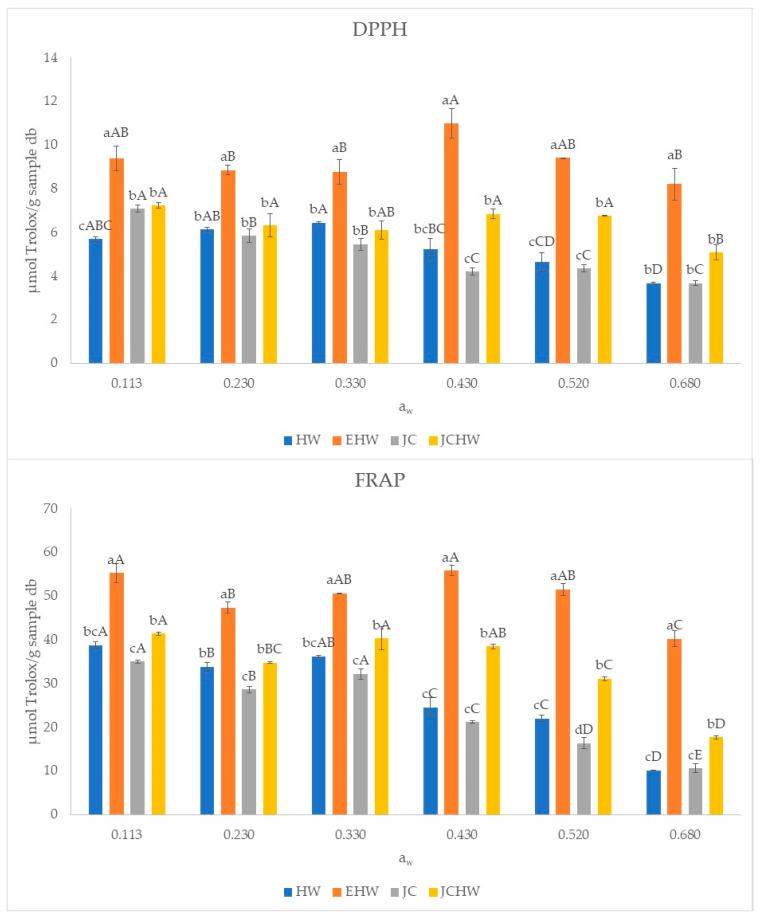
Antioxidant capacity of the samples (DPPH and FRAP) at different water activities. HW: hot water; EHW: extrusion + hot water; JC: jet cooker; JCHW: jet cooker + hot water. Different small letters for the same a_w_ indicate significant difference according to Tukey test (*p* < 0.05) between treatments. Different capital letters in the same treatment indicate significant difference according to Tukey test (*p* < 0.05) between water activities.

**Table 1 foods-11-03615-t001:** Sorption isotherms parameters. HW: hot water; EHW: extrusion + hot water; JC: jet cooker; JCHW: jet cooker + hot water.

		HW	EHW	JC	JCHW
**BET**	w_o_ (g water/g solids)	0.121	0.098	0.158	0.094
	C	1.240	1.785	0.756	2.356
	R^2^	0.976	0.915	0.811	0.947
**Henderson**	f	−1.519	−1.571	−1.440	−1.666
	n	0.895	0.952	0.833	1.012
	R^2^	0.997	0.996	0.997	0.995
**Caurie**	r	72.509	56.746	97.413	45.045
	w_s_ (g water/g solids)	0.053	0.054	0.050	0.056
	R^2^	0.962	0.970	0.953	0.975

**Table 2 foods-11-03615-t002:** Gordon and Taylor model parameters and critical values of water activity and water content (g water/g product) related to glass transition at 5 and 20 °C. HW: hot water; EHW: extrusion + hot water; JC: jet cooker; JCHW: jet cooker + hot water.

	Gordon and Taylor Model			Critical Values at 20 °C		Critical Values at 5 °C	
	T_g_ (as)	k	R^2^	CWC	CWA	CWC	CWA
**HW**	26.57	3.48	0.986	0.0120	0.080	0.0424	0.232
**EHW**	38.36	3.64	0.970	0.0315	0.173	0.0614	0.309
**JC**	19.43	3.01	0.989	-	-	0.0331	0.208
**JCHW**	29.40	3.52	0.988	0.0169	0.082	0.0471	0.222

Critical values were calculated at 20 °C and 5 °C in all cases. At 20 °C, it was not possible to determine these for JC.

**Table 3 foods-11-03615-t003:** Colour attributes and colour differences (ΔE* was calculated as the difference with the colour at a_w_ = 0.113). HW: hot water; EHW: extrusion + hot water; JC: jet cooker; JCHW: jet cooker + hot water.

		0.113	0.230	0.330	0.430	0.520	0.680
L*	HW	82.39 ± 0.14 ^Aa^	81.09 ± 0.21 ^Aa^	77.02 ± 0.20 ^Ba^	61.88 ± 0.84 ^Cb^	39.83 ± 0.27 ^Db^	34.73 ± 0.83 ^Eb^
	EHW	74.74 ± 0.04 ^Ad^	72.98 ± 0.24 ^Bc^	69.33 ± 0.17 ^Cc^	61.81 ± 0.56 ^Db^	54.64 ± 0.76 ^Ea^	39.54 ± 0.71 ^Fa^
	JC	80.16 ± 0.05 ^Ac^	79.38 ± 0.14 ^Ab^	66.98 ± 0.67 ^Bd^	54.51 ± 0.64 ^Cc^	36.23 ± 0.45 ^Dc^	32.54 ± 0.41 ^Ec^
	JCHW	81.18 ± 0.12 ^Ab^	79.43 ± 0.20 ^Ab^	75.66 ± 0.30 ^Bb^	68.36 ± 0.31 ^Cd^	53.98 ± 1.69 ^Da^	35.57 ± 1.19 ^Eb^
C*	HW	20.90 ± 0.14 ^Dc^	22.07 ± 0.13 ^Cc^	25.06 ± 0.03 ^Bb^	26.32 ± 0.18 ^Aa^	20.54 ± 0.18 ^Dc^	18.56 ± 0.45 ^Ea^
	EHW	22.58 ± 0.05 ^Da^	23.43 ± 0.17 ^Ca^	25.45 ± 0.06 ^Ab^	24.97 ± 0.41 ^ABb^	24.61 ± 0.24 ^Ba^	17.24 ± 0.15 ^Eb^
	JC	22.13 ± 0.06 ^Cb^	22.78 ± 0.02 ^Cb^	26.35 ± 0.43 ^Aa^	24.59 ± 0.55 ^Bbc^	19.38 ± 0.32 ^Dd^	16.60 ± 0.55 ^Eb^
	JCHW	20.00 ± 0.10 ^Cd^	21.09 ± 0.10 ^Cd^	23.37 ± 0.06 ^ABc^	24.15 ± 0.16 ^Ac^	22.79 ± 0.54 ^Bb^	15.77 ± 0.70 ^Ec^
h*	HW	91.61 ± 0.08 ^Ab^	90.25 ± 0.12 ^Ab^	87.87 ± 0.13 ^Ba^	82.09 ± 0.48 ^Cb^	70.95 ± 0.72 ^Dc^	68.79 ± 1.25 ^Ea^
	EHW	81.82 ± 0.11 ^Ad^	83.71 ± 0.12 ^Bd^	81.61 ± 0.23 ^Cc^	78.44 ± 0.07 ^Dc^	74.47 ± 0.23 ^Eb^	66.27 ± 0.24 ^Fb^
	JC	90.64 ± 0.04 ^Ac^	89.73 ± 0.04 ^Ac^	83.96 ± 0.37 ^Bb^	77.98 ± 0.85 ^Cc^	68.13 ± 0.68 ^Dd^	66.84 ± 0.20 ^Eb^
	JCHW	91.81 ± 0.12 ^Aa^	90.63 ± 0.04 ^Ba^	88.10 ± 0.31 ^Ca^	84.26 ± 0.33 ^Da^	77.85 ± 0.13 ^Ea^	67.18 ± 0.43 ^Fb^
ΔE*	HW	-	1.84 ± 0.24 ^Ea^	6.97 ± 0.15 ^Db^	21.57 ± 0.87 ^Cb^	43.21 ± 0.30 ^Ba^	48.35 ± 0.89 ^Ab^
	EHW	-	2.02 ± 0.18 ^Ea^	6.28 ± 0.16 ^Dc^	13.42 ± 0.47 ^Cc^	20.65 ± 0.74 ^Bc^	36.17 ± 0.69 ^Ac^
	JC	-	1.08 ± 0.10 ^Eb^	14.10 ± 0.46 ^Da^	26.27 ± 0.63 ^Ca^	44.75 ± 0.49 ^Ba^	48.58 ± 0.44 ^Aa^
	JCHW	-	2.11 ± 0.22 ^Ea^	6.62 ± 0.30 ^Db^	13.72 ± 0.43 ^Cc^	27.84 ± 1.60 ^Bb^	46.50 ± 1.15 ^Ab^

Different small letters in the same parameter (L*, C*, h* and ΔE*) for the same a_w_ indicate significant difference according to Tukey test (*p* < 0.05) between treatments. Different capital letters in the same row indicate significant difference according to Tukey test (*p* < 0.05) between water activities.

## Data Availability

The datasets generated for this study are available on request to the corresponding author.

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
