# Peer review of "Sorption Isotherms, Glass Transition and Bioactive Compounds of Ingredients Enriched with Soluble Fibre from Orange Pomace"

_foods, 2022, doi:10.3390/foods11223615_

Round 1

Reviewer 1 Report

The Manuscript "Sorption isotherms, glass transition and bioactive compounds of ingredients enriched with soluble fibre from orange pomace" provides the behaviour and the physicochemical proprierties of fibers obtained from citrus by-products in detail.

I reccomended to authors to improve the Section Materials and methods . In particular the description about the determination of antiosxidant capacity (Folin, DPPH..). 

Authors should better present the data in the table (double check the journal's indications)

Author Response

Reviewer #1

The Manuscript "Sorption isotherms, glass transition and bioactive compounds of ingredients enriched with soluble fibre from orange pomace" provides the behaviour and the physicochemical proprierties of fibers obtained from citrus by-products in detail.

I reccomended to authors to improve the Section Materials and methods. In particular the description about the determination of antiosxidant capacity (Folin, DPPH..). 

This was described in lines 161-174.  

Authors should better present the data in the table (double check the journal's indications) –

Journal’s indications were checked and headings of tables were improved with the addition of the acronyms explanation.  

Reviewer 2 Report

Keywords completely unrelated to the job title

Explanation explaining how to take care of safety, safety from a safety point of view, and to improve quality and safety to safety. The authors understand and are able to manage the size and information of the isotherm changes in each of its segments. The advantage of the work is the use of isotherms, not only the top, fast, without extending the physicochemical knowledge - active water.

line 74 - it is necessary to provide the water temperature, drinking "hot water" in analytical work is very imprecise

line 150 - it is very inappropriate to describe "The samples were centrifuged .." is the correct name of the device, not of processes, physicochemistry.

I have no major comments as to the palette of the experiment, its execution and discussion. It seems to me, however, that the conclusions are not too closely related to the results of the work. The authors will bring in the conclusions something reminiscent of discussions, not a summary - what turned out, what is the most important.

Author Response

Reviewer #2

Keywords completely unrelated to the job title –

Keywords were reviewed. Some words were changed and the order was rearranged.

Explanation explaining how to take care of safety, safety from a safety point of view, and to improve quality and safety to safety. The authors understand and are able to manage the size and information of the isotherm changes in each of its segments. The advantage of the work is the use of isotherms, not only the top, fast, without extending the physicochemical knowledge - active water.

line 74 - it is necessary to provide the water temperature, drinking "hot water" in analytical work is very imprecise

This was described later in the paper, lines 86, 90 and 94.  

line 150 - it is very inappropriate to describe "The samples were centrifuged .." is the correct name of the device, not of processes, physicochemistry.

Centrifuge model was added in line 156.

I have no major comments as to the palette of the experiment, its execution and discussion. It seems to me, however, that the conclusions are not too closely related to the results of the work. The authors will bring in the conclusions something reminiscent of discussions, not a summary - what turned out, what is the most important.

Conclusions were reviewed and improved (lines 396 – 400).

Reviewer 3 Report

This paper is focused on the sorption behaviour, glass transition, mechanical properties, color and bioactives of four different soluble fibre enriched powders obtained from orange pomace using green technologies.

Although the manuscript is well presented,  I suggested to the authors to introduce some minor changes to improve the paper. The authors need to provide or add a wider description of other technologies for obtaining fibre enriched ingredient which can be viewed from several aspects such as economic, environmental, practicality, and several other important aspects.

The authors needs to confirm and clarify about replication for this research. If this research has been replicated, the author needs to provide or add information about the number of replications that have been carried out.

Specific comments:

- 2.5 Caption to correct, it shoud be Colour analysis

Author Response

Reviewer #3

This paper is focused on the sorption behaviour, glass transition, mechanical properties, color and bioactives of four different soluble fibre enriched powders obtained from orange pomace using green technologies.

Although the manuscript is well presented,  I suggested to the authors to introduce some minor changes to improve the paper. The authors need to provide or add a wider description of other technologies for obtaining fibre enriched ingredient which can be viewed from several aspects such as economic, environmental, practicality, and several other important aspects.

 A more thorough description of the possible uses of the orange pomace was added, highlighting the importance of the use of “green technologies” without the use of solvents (lines 36-42).  

The authors needs to confirm and clarify about replication for this research. If this research has been replicated, the author needs to provide or add information about the number of replications that have been carried out.

Replications carried out were added in each experiment (lines 140, 174).

Specific comments:

- 2.5 Caption to correct, it shoud be Colour analysis –

 Manuscript was corrected.